REGISTERED REPORT PROTOCOL

# Utility of the monocyte to lymphocyte ratio in diagnosing latent tuberculosis among HIV-infected individuals with a negative tuberculosis symptom screen

**Jonathan Mayito**[1]*, **David B. Meya**[1¤], **Joshua Rhein**[2], **Christine Sekaggya-Wiltshire**[1]

**1** Infectious Diseases Institute, Makerere University College of Health Sciences, Kampala, Uganda,
**2** Division of Infectious Diseases and International Medicine, University of Minnesota, USA Department of Research, Minneapolis, Minnesota, United States of America

¤ Current address: Department of Medicine, College of Health Sciences, Makerere University, Kampala, Uganda
* jmayito@idi.co.ug

This is a Registered Report and may have an associated publication; please check the article page on the journal site for any related articles.

## Abstract

### Background

Latent Tuberculosis Infection (LTBI) remains a major driver of the TB epidemic, and individuals with Human Immuno-deficiency Virus (HIV) are particularly at a heightened risk of developing LTBI. However, LTBI screening among HIV-infected individuals in resource limited setting is largely based on a negative symptom screen, which has low specificity.

### Methods

In a cross sectional diagnostic study, 115 HIV infected participants with a negative symptom screen will be consented and enrolled. They will be requested to donate 5 ml of blood for complete blood count (CBC) and interferon gamma release assay (IGRA) testing. In a nested prospective study, the 115 participants will be initiated on Tuberculosis Preventive Therapy and the CBC testing repeated after 3 months. In the analysis of study finding, the monocyte to lymphocyte ratio (MLR) will be derived from the dividend of the absolute monocyte and lymphocyte counts. The optimal MLR positivity cut-off for elevated or normal MLR will be the highest value of Youden's index, J (sensitivity + specificity-1). The MLR will be cross tabulated with the IGRA status to determine the sensitivity, specificity, negative and positive predictive values of the MLR. The area under the receiver operating characteristic (ROC) curve will be determined to give the overall diagnostic accuracy of MLR. The baseline and 3 month CBC will be used to determine the change in MLR, and a random effect logistic regression will be used to determine factors associated with the change in the MLR.

### Discussion

If positive results are realized from this study, the MLR could become an inexpensive alternative biomarker with potential to improve the specificity of the negative symptom screen in identifying individuals that should be targeted for TB preventive therapy.

**Data Availability Statement:** All relevant data from this study will be made available upon study completion.

**Funding:** Research reported in this publication was supported by the Fogarty International Center of the National Institutes of Health under grant #D43TW009345 awarded to the Northern Pacific Global Health Fellows Program. DELTAS Africa Initiative (ref. DEL-15-011, via THRiVE-2) funds JM's PhD studies. The DELTAS Africa Initiative is an independent funding scheme of the AAS's Alliance for Accelerating Excellence in Science in Africa and supported by the NEPAD Agency with funding from the Wellcome Trust Grant No. 107742/Z/15/Z and the United Kingdom government. The content is solely the responsibility of the authors and does not necessarily represent the official views of the National Institutes of Health. The funders had and will not have a role in study design, data collection and analysis, decision to publish, or preparation of the manuscript.

**Competing interests:** The authors have declared that no competing interests exist.

# Background

Latent Tuberculosis Infection (LTBI) remains a major driver of the tuberculosis (TB) epidemic. Up to 1.7 billion people worldwide are latently infected, and at risk of TB disease [1]. The Human Immuno-deficiency Virus (HIV) positive individuals are 20–30 times more likely to develop LTBI, and have a 5–15% risk per year and a 30% life time risk of developing active TB [2, 3]. The World Health Organization (WHO) END Tuberculosis (TB) Strategy aims to end the epidemic by 2030, partly through scaling up screening of TB contacts and high risk groups for LTBI treatment [4].

However, LTBI diagnosis is still a challenge; the Interferon Gamma Release Assay (IGRA) and Tuberculin Skin Test (TST) on which LTBI diagnosis is based, are memory T-cell response based tests, which may remain positive beyond the clearance of the infection [5], making identification of those at ultimate risk of progressing to active TB challenging. Furthermore, IGRA use is limited by; cost, requirement for special equipment and personnel, while TST use is limited by; false positives due to BCG vaccination and cross-reaction with environmental non-tuberculous Mycobacteria, need for two patient visits, and false negative results due to; anergy, malnutrition, immuno-suppression and incorrect administration [6, 7].

Currently, WHO recommends an LTBI screening algorithm based on a negative symptom screen including; cough, fever, night sweats and weight loss, to identify HIV infected patients eligible for Tuberculosis Preventive Therapy (TPT). Because of the limitations mentioned earlier and low positive predictive values (IGRA—2.7% vs TST—1.5–2.1%) for progression to active TB, IGRA and TST are recommended as additional tests where feasible [8–10]. This presents two potential problems; 1) using an approach with limited specificity in identifying those at ultimate risk of LTBI, making it expensive and also catching individuals who may not need preventive treatment, and 2) misdiagnosis of subclinical TB as LTBI, consequently targeting it with inappropriate therapy.

The Monocyte to Lymphocyte Ratio (MLR), is a rapid and inexpensive biomarker with potential to differentiate LTBI from active TB, because a higher MLR occurs in adults with active TB (0.5 IQR [0.36–0.64]) compared to those with LTBI (0.25 IQR [0.20–0.28]) [11]. Similarly, in children with confirmed TB (0.47 IQR [0.38–0.68]) compared to those unlikely to have TB (0.21 IQR [0.14–0.39] [12]. An MLR cut-off value of 0.38 in HIV-infected Kenyan children had a sensitivity of 77%, specificity of 78%, positive predictive value of 24%, and negative predictive value of 97% for identification of active TB [12]. A lower cut-off (0.29) in HIV-uninfected Italians had high sensitivity (91%) and specificity (94%) [12]. In addition to being useful for diagnosis, the MLR can also monitor TB treatment because it declines with therapy, before (0.41 [IQR: 0.38–0.68]) and after (0.11 [IQR: 0.048–0.348]) TB treatment [13]. Finally, an MLR greater than 0.87 was associated with an increased risk of active TB within 5 years of anti-retroviral therapy (ARVs) initiation [11, 14].

This study will therefore evaluate the diagnostic performance of the MLR compared to IGRA, and measure the change in MLR from baseline to three months during TPT, among HIV-infected patients with a negative WHO symptom screen. The MLR has potential to become an alternative biomarker for screening for LTBI and identifying individuals more likely to benefit from TPT. This would help in targeting preventive therapy to only individuals at utmost risk for active TB, given the enormous task it would be to reach all the 1.7 billion people estimated to be infected. Targeting preventive treatment to those at utmost risk would not only be cost effective but also more feasible, particularly in resource limited settings. Positive results from this study, will therefore provide valuable information on MLR as a potential biomarker to bridge gaps in LTBI diagnosis.

The study primary objective is as follows: To determine the diagnostic performance of the monocyte to lymphocyte ratio against IGRA in diagnosing latent TB among HIV-infected individuals with a negative WHO TB symptom screen.

And the secondary Study Objective is as follows: To determine the change in the monocytes to lymphocytes ratio measured at baseline, and three months among HIV-infected individuals during tuberculosis preventive therapy.

The hypothesis underpinning the study objectives is grounded in the relationship of *M.tb* with monocytes as the target cells for infection, and lymphocytes as the main immune effectors cells against *M.tb* infection. We therefore hypothesize that *M.tb* through the interferon gamma signaling effect on hematopoiesis leads to proliferation of the myeloid biased hematopoietic stem cells (HSCs) to produce target cells for propagating infection relative to the lymphoid biased HSCs leading to a higher MLR in TB infection.

## Study methods

### Study design

Primary objective: A cross sectional diagnostic study will be carried out among HIV infected patients to determine the diagnostic performance of the MLR against IGRA in diagnosing LTBI.

Secondary objective: A nested prospective study will be carried out among HIV infected individuals to determine the change in the MLR measured at baseline, and after three months of TPT.

### Study participants

**Study eligibility criteria.** The study participants will be HIV infected patients with a negative TB symptom screen regardless of whether they have initiated anti-retroviral therapy (ART) or not; with no clinical (negative WHO TB four symptom screen) symptoms of active TB, aged 18 years and above, able to give written informed consent to participate in the study, and are contactable by telephone to allow communication to ensure attendance of the follow up visit.

On the other hand, the study will exclude anyone; who has ever received TPT or standard anti-mycobacterial treatment, or has a known medical condition which may interfere with the participant's ability to participate in the study such as blood dyscrasia; sickle cell anemia, myelodysplastic syndromes, leukemia, or any current viral or bacterial infections.

**Participant recruitment.** The participants will be recruited and followed up from the Infectious Disease Institute (IDI) which is an urban HIV out-patient clinic with over eight thousand HIV patients under its care. Participants will also be recruited from Kampala City Council Authority (KCCA) health centers, which will act as only recruitment sites. The study participants will be consecutively approached, briefed about the study, and interested participants consented for screening according to the study eligibility criteria as described above. Eligible participants who give written informed consent will be screened and recruited into the study.

### Test methods

**Index test.** The index test will be the MLR derived from dividing the absolute monocyte count by the absolute lymphocyte count. The absolute counts will be obtained from a complete blood count (CBC) that will be performed using a Beckman Coulter Ac•T 5diff AL (Autoloader) Hematology Analyzer. The optimal MLR positivity cut-off for elevated or normal MLR will be the highest value of Youden's index, J (sensitivity + specificity-1). Values greater than the optimal diagnostic cut-off will indicate LTBI while those below imply that LTBI is unlikely. A receiver operating characteristic (ROC) curve derived from sensitivity and specificity of the

MLR will be generated, and the area under the curve will give the overall diagnostic power of the MLR.

**Reference test.** The reference test will be the IGRA, particularly **QuantiFERON®-TB Gold Plus** (QFT-**Plus**) by Qiagen. The IGRA is currently the best available test for the diagnosis of LTBI due to lack of a gold standard. It is a commercial qualitative ELISA assay where 1 ml of blood is collected in each of the four tubes containing synthetic peptides; ESAT-6 (green)–TB1 and CFP-10 (yellow top)–TB2, Phytohaemagglutinin (PHA) (purple top)—Mitogen as the positive control and no antigen in the fourth tube (gray top)—Nil as the negative control. The positive, negative and intermediate results are determined as shown in the Table 1 below (QuantiFERON—TB Gold Plus Results table version 2.71).

The results of the index test, reference test, and the clinical information will not be available to the laboratory staff at the time the tests assessment is carried out. The CBC and results of IGRA are automatically generated from the Coulter and ELISA reader machines respectively, and therefore will not be influenced by operator staff bias or the clinician.

## Ethical consideration

The study protocol, informed consent forms, case report forms, and recruitment materials were reviewed and approved (SBS-794) on 24/06/2020 by the Makerere University College of Health Sciences' School of Bio-medical Sciences Ethics Review Committee and now in the process of being submitted to the Uganda National Council of Science and Technology (UNSCT), prior to implementation of any study related activities. All participants will give informed consent prior to participation, and confidentiality will be ensured by using subject identification numbers, only authorized study staff having access to study documents and all study documents being stored under a double lock storage system.

## Study visits

**Study evaluations.** *Clinical Procedures*. The general flow of the study procedures is illustrated in Fig 1.

*Baseline visit*. HIV-infected individuals willing to participate in the study will be assessed for; cough, weight loss, fevers and night sweats. Participants with a positive screen (at least one symptom present) will be referred for further TB screening while those with a negative screen will be screened for the study. In addition, a thorough physical examination for further TB screening will be carried out including; respiratory examination, and systemic assessment for clinical symptoms suggestive of TB, for example; lymphadenopathy, abdominal masses, and vertebral column abnormalities. Participants with a negative symptom screen and no clinical findings suggestive of TB will be consented for enrollment in the study. They will then be

**Table 1. Interpretation of QuantiFERON—TB Gold Plus Results.**

| Nil (IU/mL) | TB1 minus Nil (IU/mL) | TB2 minus Nil (IU/mL) | Mitogen minus Nil (IU/mL) | QFT-Plus Result | Report/Interpretation |
|---|---|---|---|---|---|
| ≤ 8.0 | ≥ 0.35 and ≥ 25% of Nil | Any | Any | Positive | *M. tuberculosis* infection likely |
| | Any | ≥ 0.35 and ≥ 25% of Nil | | | |
| | <0.35 OR ≥ 0.35 and < 25% of Nil | | ≥ 0.5 | Negative | *M. tuberculosis* NOT infection likely |
| | | | < 0.5 | Intermediate | Likelihood of M. tuberculosis infection cannot be determined |
| < 0.8 | Any | | | | |

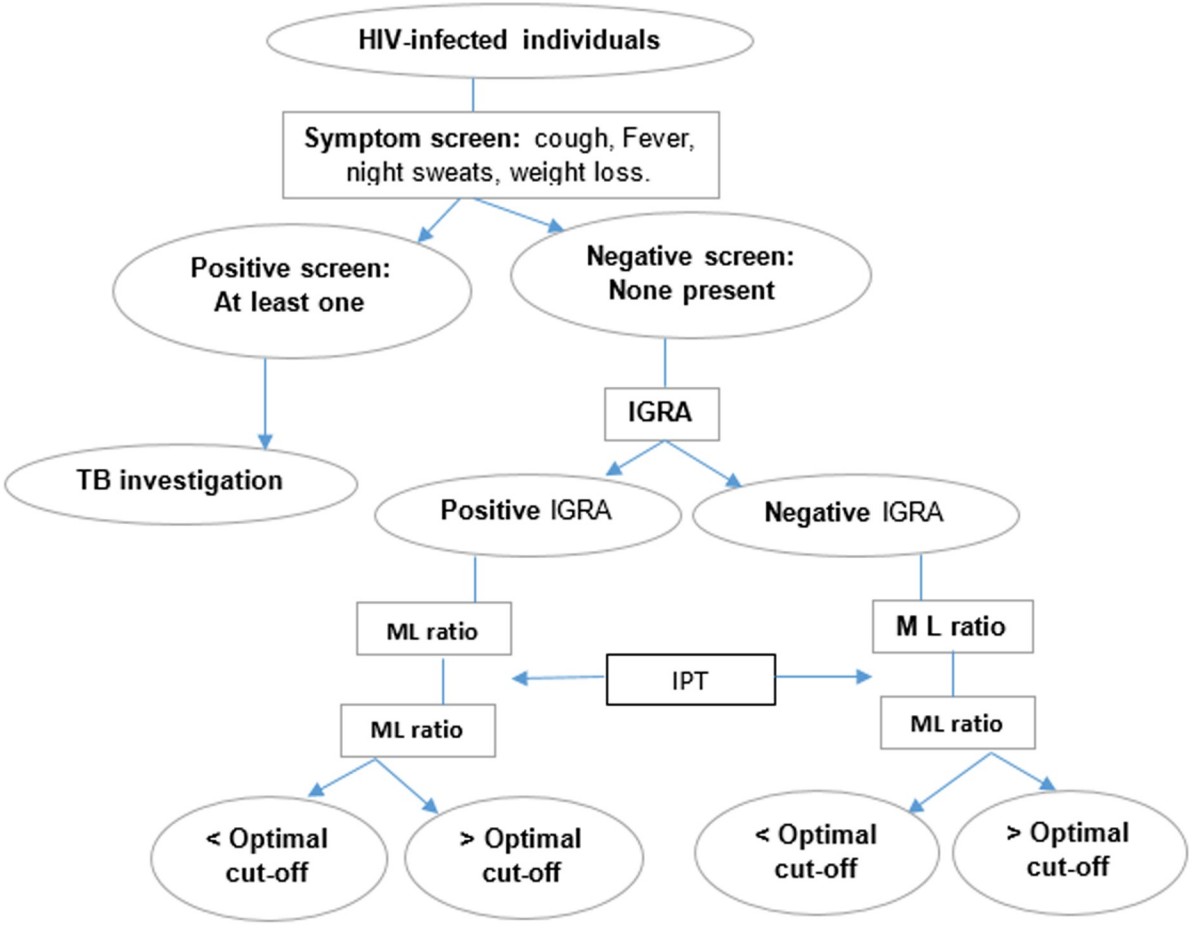

**Fig 1. Study flow chart.**

administered the study questionnaire and asked to donate 5 ml of whole blood for IGRA (4 ml) and CBC (1 ml) testing. These are illustrated in *Table 2*.

*Follow up visits.* The enrolled participants will then be initiated on the currently recommended TPT regimen i.e. isoniazid 300 mg daily for 6 months or 3HP (weekly isoniazid rifapentine for 3 months), and CBC measurement repeated after 3 months of treatment. Adherence support will be provided through the next of kin for daily reminders and monthly phone call reminders. The participants will also be linked to HIV care services if they are not already in care. Participants will be encouraged to seek medical care at any health facility or contact the study team about adverse events between scheduled study visits. The follow up procedures are also illustrated in *Table 2*.

## Laboratory procedures

**Specimen collection.** Blood for IGRA testing will be collected in the special quantiFERON tubes and transported to the laboratory at room temperature within 16 hours from collection. The blood for CBC testing will be collected in EDTA vacutainer tubes. The IGRA

**Table 2. Schedule of events.**

| Protocol activity | Baseline | Month 1 | Month 2 | Month 3 |
|---|---|---|---|---|
| Consent | x | | | |
| History and examination | x | | | x |
| Questionnaire | x | | | x |
| IGRA | x | | | |
| CBC | x | | | x |
| Phone call/adherence | x | x | x | x |

test and CBC will be performed in the IDI Translation Research Laboratory and the IDI Makerere University-John Hopkins University (MUJHU) laboratory respectively.

*Specimen preparation, handling and storage.* The IGRA samples will be incubated at 37˚C for 16 to 24 hours, centrifuged at 3000g for 10 minutes and the serum stored at -80˚C until analysis. The analysis will follow the standard operating procedure as provided by the Qiagen Company, the makers of the QuantiFERON kits. The CBC will be analyzed immediately or stored at 4˚C till analysis using the using the Beckman Coulter Ac•T 5diff AL (Autoloader) Hematology Analyzer.

## Statistical methods

### Sample size and power

Sample size was derived using the formula for diagnostic studies based on sensitivity;

$n = \frac{Z^2_{1-\alpha} \times S_N \times (1-S_N)}{L^2 \times prevalence}$ [15], where $S_N$—anticipated sensitivity, was set at 77% [12], and $L^2$—absolute precision required, set at 10%. Using an LTBI ***Prevalence*** of 65% (based on QuantiFERON) around Kampala suburbs [16], and considering a 10% loss to follow up, a sample size of **115 participants** will give an 80% power of delineating the diagnostic performance of MLR.

### Analysis of endpoints

The Independent variables will include; socio-demographics, body mass index (BMI), BCG status, HIV viral load, ART status, smoking status, alcohol intake, intensity and duration of contact with TB patient (where there is known contact with a TB patient), and inter-current infections between study visits.

### Descriptive analysis

The continuous variables for example age, weight and height will be reported as means with their standard deviations for normally distributed variables, while the median will be reported for skewed distribution since it is more stable to outliers. The categorical variables for example BMI, BCG status, viral load (detected or not detected), ART status, smoking, alcohol intake, and time of contact will be reported as proportions in terms of frequencies and percentages.

### Diagnostic accuracy of MLR

The MLR will be derived as a dividend of the absolute monocyte count by the lymphocyte absolute count. The optimal MLR positivity cut-off for elevated or normal MLR will be the highest value of Youden's index, J (sensitivity + specificity-1) derived from the sensitivity and specificity of all possible cut-off values on the ROC curve. The sensitivity, specificity, positive and negative predictive values will be determined from a 2X2 table between MLR and IGRA, and the diagnostic power of the MLR determined from area under the ROC curve.

## Change in MLR

The change in MLR will be determined by subtracting MLR at 3 months from baseline MLR. Patients will be categorized as IGRA positive and negative, and the change in MLR will be compared between IGRA-positive and IGRA-negative using a paired T-test for paired proportions. A random effect logistic regression analysis will be conducted to determine factors associated with the change in the MLR. This model will cater for the variation arising from loss of independence due to the repeated measurement.

Confounders: Patients who are known to have conditions that can alter the MLR such as blood dyscrasia; sickle cell anemia, myelodysplastic syndromes, leukemia, or any current viral or bacterial infections will be excluded from the study at enrolment. Any acute bacterial or viral infections occurring during follow up will be noted and included in the regression model to see if they had any impact.

## Dissemination of study results

Dissemination of the results of this study will be done through multiple methods including; publication in a peer-reviewed journal and presentation at an international conference with associated media coverage. The results will also be availed to the National TB and Leprosy Program at Ministry of Health, IDI research department, study participants and other policy stakeholders. The results will also be made available as both printed and online resources. The sponsors will also receive periodic reports. The results will be presented in a clear and concise format appropriate to the target audience such as journal article for research scientists, media briefs for media and study participants' briefs for the study participants.

## Discussion

The MLR is derived from the CBC, a test that is commonly carried out in clinical setting, and therefore would easily be scale up in resource setting. Apart from TB, the MLR as an inflammatory marker has shown usefulness in determining prognosis of certain disease conditions including cancer, rheumatoid arthritis and coronary artery disease [17–19]. Its potential in differentiating between LTBI and active TB has been highlighted in the background above. However, its role in identifying asymptomatic individuals with LTBI who would benefit from TPT has not been evaluated, and this is the focus of this study.

The study has been designed to evaluate the MLR against IGRA, to give an impression of the diagnostic utility of the MLR. This will provide useful information about the MLR as an alternative biomarker that could improve the currently used symptom screen in identifying individuals to be targeted for TPT. Improving the current screening algorithm would reduce the number of individuals exposed to treatment that may not benefit to them and those with sub-clinical TB disease inappropriately treated with TPT. According to prevalence data in Uganda, up to; 51 percent, and 35 percent of individuals, would not require TPT based on TST and IGRA testing respectively [20, 21].

Furthermore, this study by measuring change in MLR during TPT therapy, will provide information on the MLR's ability to monitor response to TPT. Currently, there is lack of a laboratory or radiological test to monitor response to TPT, partly because the IGRA and TST have low reversion rates. The current practice is that individuals are initiated on TPT but at the end of the preventive therapy, we are unable to tell who has cleared or failed to clear the infection, and therefore in need of prolonged or alternative therapy. The MLR as highlighted earlier has been shown to decline with treatment of active TB and could be of value in determining the success of the preventive therapy.

This study therefore seeks to address the above two critical gaps in the diagnosis of LTBI and monitoring response to its treatment, which gaps have driven the global efforts to control TB towards massive treatment. Massive treatment of potential risk groups is not only tedious but is also not cost effective, and therefore less attractive to low resource settings which bear the largest burden of LTBI. This study will therefore evaluate MLR as a biomarker that may improve the risk stratification to better target TPT to those at utmost risk.

Our study has some limitations. Due to lack of a gold standard for the diagnosis of LTBI, we will use IGRA, the currently best available LTBI test, as the reference test for the MLR. The MLR is also a non-specific biomarker and like other parameters of CBC, may be affected by concurrent infections, malignancies and blood dyscrasias, which may limit its use in individuals with these conditions, and as such would more likely be deployed as a screening biomarker. The study will make all efforts to exclude individuals known to have these conditions at baseline, while the infections that occur during follow up will be included in the regression analysis to determine if they affected the outcome. Another limitation is that the study may not be able to recruit enough patients not ARVs versus those on ARVs so as to allow studying the effect of ARVs on the MLR.

In summary, this study will determine the diagnostic utility of the MLR for LTBI and monitoring response to preventive therapy. It is hoped that the results, will present the MLR as an inexpensive alternative biomarker that will improve the specificity of the symptom screen in identifying individuals that should be targeted for preventive therapy.

## Adverse event reporting

At the follow up visits, the study investigator will assess for adverse events (AEs) that may have occurred since the previous visit. The investigators will generate and submit annual reports summarizing these AEs to the institutional review board and UNCST. All AEs will be managed according to the standard clinical protocol of IDI and Mulago national referral hospital.

## Supporting information

**S1 File.**
(PDF)

## Author Contributions

**Conceptualization:** Jonathan Mayito, Christine Sekaggya-Wiltshire.

**Data curation:** Jonathan Mayito.

**Formal analysis:** Jonathan Mayito.

**Funding acquisition:** Jonathan Mayito, David B. Meya, Joshua Rhein, Christine Sekaggya-Wiltshire.

**Methodology:** Jonathan Mayito, Christine Sekaggya-Wiltshire.

**Project administration:** Jonathan Mayito.

**Supervision:** David B. Meya, Joshua Rhein, Christine Sekaggya-Wiltshire.

**Validation:** Jonathan Mayito.

**Visualization:** Jonathan Mayito.

**Writing – original draft:** Jonathan Mayito.

**Writing – review & editing:** Jonathan Mayito, David B. Meya, Joshua Rhein, Christine Sekaggya-Wiltshire.

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
