## [Decision Letter · Decision Letter 0]

9 Sep 2020

PONE-D-20-22616

Utility of the monocyte to lymphocyte ratio in diagnosing latent tuberculosis among HIV-infected individuals with a negative tuberculosis symptom screen

PLOS ONE

Dear Dr. Mayito,

Thank you for submitting your manuscript to PLOS ONE. After careful consideration, we feel that it has merit but does not fully meet PLOS ONE’s publication criteria as it currently stands. Therefore, we invite you to submit a revised version of the manuscript that addresses the points raised during the review process.

We look forward to receiving your revised manuscript.

Kind regards,

Hasnain Seyed Ehtesham

Academic Editor

PLOS ONE

Journal Requirements:

'The study protocol, informed consent forms, case report forms, and recruitment materials will be reviewed and approved by the Makerere University College of Health Sciences’ School of Bio-medical Sciences Ethics Review Committee and the Uganda National Council of Science and Technology (UNSCT), prior to implementation of any study related activities.'  

(a) Please amend your current ethics statement to state that the proposed study was been reviewed and approved by the ethics committee.

(b) Once you have amended this/these statement(s) in the Methods section of the manuscript, please add the same text to the “Ethics Statement” field of the submission form (via “Edit Submission”).

For additional information about PLOS ONE ethical requirements for human subjects research, please refer to " ext-link-type="uri" xlink:type="simple">http://journals.plos.org/plosone/s/submission-guidelines#loc-human-subjects-research."

3. In your Methods section, please provide additional information about the participant recruitment method and the demographic details of your participants. Please ensure you have provided sufficient details to replicate the analyses such as:  a) a description of how participants will be recruited and b) descriptions of where participants will be recruited and where the research will take place.

5. We note you have included a table to which you do not refer in the text of your manuscript. Please ensure that you refer to Table 2 in your text; if accepted, production will need this reference to link the reader to the Table.

6. Your ethics statement must appear in the Methods section of your manuscript. If your ethics statement is written in any section besides the Methods, please move it to the Methods section and delete it from any other section. Please also ensure that your ethics statement is included in your manuscript, as the ethics section of your online submission will not be published alongside your manuscript.

Additional Editor Comments (if provided):

Major Revision

Reviewers' comments:

Reviewer's Responses to Questions

**Comments to the Author**

1. Does the manuscript provide a valid rationale for the proposed study, with clearly identified and justified research questions?

Reviewer #1: Yes

Reviewer #2: Yes

Reviewer #3: Partly

2. Is the protocol technically sound and planned in a manner that will lead to a meaningful outcome and allow testing the stated hypotheses?

Reviewer #1: Partly

Reviewer #2: Partly

Reviewer #3: Partly

3. Is the methodology feasible and described in sufficient detail to allow the work to be replicable?

Reviewer #1: No

Reviewer #2: Yes

Reviewer #3: Yes

4. Have the authors described where all data underlying the findings will be made available when the study is complete?

Reviewer #1: Yes

Reviewer #2: Yes

Reviewer #3: Yes

5. Is the manuscript presented in an intelligible fashion and written in standard English?

Reviewer #1: No

Reviewer #2: No

Reviewer #3: No

6. Review Comments to the Author

You may also provide optional suggestions and comments to authors that they might find helpful in planning their study.

Reviewer #1: Jonathan et al, in their Protocol suggest utilization of MLR and IGRA as prognostic marker to corelate the Latent TB with HIV disease severity.

Major comments / concerns

Author should justify the 115 number of patients , from which calculation they came to this study, would this be a clinical trials or prospective study

I found this proposal somewhat ambitious because author propose to use monocyte to lymphocyte ratio as prognostic marker specially for latent TB. Here , my opinion would be revisit literature and revisit the concept .

Pls revise the text pertaining to the influence of IFN on proliferation of HSC. This is fundamental error, HSC require G/M-CSF or CSF-1 for proliferation and not IFN !!

Monocyte population is just one rough parameter which is used in CBC. However for the disease management , it is macrophages which are more relevant like in granuloma and in latent TB, M2 or foamy macrophage are most relevant population which promote latency and not mere monocyte.

This is also due to fact that Both naive Monocyte and macrophages and CD14+/; F4/80+. During acute / active TB infection Monocyte get differentiated into CD11b+/ CD68+/Inos+ Th1 primed macrophages. during persistent / latent episode of TB , same Th1 / M1 populations, interestingly, get polarize toward M2 / Foamy macrophages and hide / carry bacilli to the gaseous granulomatous lesions of lung. So author should redefine this aspect.

Other aspect is with IGRA so I want to suggest author that author should focus on Th1/Th2 kits because other than IFN, TNF , IL-4/IL-6/IL-10/IL-13 are decisive host factors which can dictate the pathogenesis of acute and latent TB. So here author may include Th1/Th2 kit as parameters which would be much robust correlation specially when including HIV+ patients.

Also for MLR , the author should use other term as in the feild of hematology / immunology MLR means mix lymphocyte ratio which is not clear here. Also I would like to suggest authors to include both M1/M2 macrophages , M1/ T cells , M1/cd4, M1/ CD8 and CD4/8 ratio as additional parameters for more in depth immunomonitoring purpose . This will give protocol strength and then analysis would be more valid.

I could not follow the arguments related to Prophylactic treatment of volunteers and why only for 3 months ? why Author only given INH and why not other 1st generation drug ?. I think it would be more appropriate to include latent TB patients which are on 2nd generation TB drug including Rapamycin and Bedaquiline drugs e.g.

Did author plan to include patients with non reactive or extra pulmonary TB with PPE as well in their cohort ??

Why author did not include PPD in their analysis !!!

In the current draft, it is not clear which group of HIV patients would be included. What is the influence of Anti retro-viral therapy also contribute to IFN gamma so from that point of view, some newly diagnosed HIV patients should also be included ( subjected to IEC approval but if author can include these patients, this would be baseline data. This would also address the pattern of IGRA.

Minor points

Finally , author should revise the entire document for the language because current version is difficult to analyze, e.g. second paragraph of Background !!! Please remove high impact word from the Dissemination paragraph. The author should be realistic.

Reviewer #2: This manuscript is rejected as this is a registered report protocol and it does not provide any new information at this stage. The study is a proposed hypothesis and still requires methods to be validated. However evaluation of diagnostic performance of Monocyte to Lymphocyte ratio (MLR) in comparison to IGRA assay still needs to be documented.

Outcome of this study will provide some data and insights into the comparative analysis which could be useful for development of a biomarker for LTBI diagnosis.

Reviewer #3: In this manuscript (Registered Report Protocol) the authors proposed the diagnostic utility of the MLR (monocyte to lymphocyte ratio) in LTBI and monitoring response to preventive therapy in HIV patients.

Following are my comments on the protocol

1) In the literature available it is mentioned that Monocyte:lymphocyte ratio (M:L) has been identified as a risk factor in development of TB disease in children and those undergoing treatment for HIV in co-infected individuals. More recently, a high M:L has been shown to distinguish persons with active and latent TB from uninfected persons, and been used to predict risk of developing TB in infants. M:L has been found to reduce after treatment for HIV, and corresponded with improvement in the patient’s condition5. Therefore, in humans it appears that the M:L shows promise as an indicator of risk of developing active TB and could facilitate the targeting of preventative treatments/therapy for those who are defined as being at greater risk.

References

Naranbhai V, et al. Ratio of monocytes to lymphocytes in peripheral blood identifies adults at risk of incident tuberculosis among HIV-infected adults initiating antiretroviral therapy. J. Infect. Dis. 2014;209:500–9. doi: 10.1093/infdis/jit494.

Naranbhai V, et al. The association between the ratio of monocytes:lymphocytes at age 3 months and risk of tuberculosis (TB) in the first two years of life. BMC Med. 2014;12:120. doi: 10.1186/s12916-014-0120-7.

So, the novelty of this work is questionable. Authors are advised to revisit the literature which is full of this kind of evidences.

2) Although monocyte to lymphocyte ratio is a good indicator of risk of developing active TB but role of macrophages also could not be ignored. It is advised to include macrophage variations also in this protocol.

3) Since HIV infection also manipulate the M/L ratio, authors should clarify what would be there control group.

References

The Journal of Infectious Diseases Vol. 162, No. 6 (Dec., 1990), pp. 1239-1244 Analysis of Lymphocytes, Monocytes, and Neutrophils from Human Immunodeficiency Virus (HlV)-Infected Persons for HIV DNA

J Infect Dis. 2014 Feb 15; 209(4): 500–509. Ratio of Monocytes to Lymphocytes in Peripheral Blood Identifies Adults at Risk of Incident Tuberculosis Among HIV-Infected Adults Initiating Antiretroviral Therapy

J Acquir Immune Defic Syndr. 2014 Dec 15; 67(5): 573–575. The association between the ratio of monocytes:lymphocytes and risk of tuberculosis(TB) amongst HIV infected postpartum women

4) The authors should also clarify the HIV patients selected for the study are on ARV therapy and if these drugs are having impact on M/L ratio.

In my opinion the study need to be redesigned keeping the above mentioned points in centre.

7. PLOS authors have the option to publish the peer review history of their article (what does this mean?). If published, this will include your full peer review and any attached files.

Reviewer #1: No

Reviewer #2: No

Reviewer #3: No

---

## [Author Response · Author response to Decision Letter 0]

15 Oct 2020

Response to reviewer comments for “Utility of the monocyte to lymphocyte ratio in diagnosing latent tuberculosis among HIV-infected individuals with a negative tuberculosis symptom screen”

Dear Senior Editor,

Thank you for inviting revision of our manuscript. Please find below responses to the queries and comments raised at the initial review of our manuscript.

Qn. We note that you have provided funding information that is not currently declared in your Funding Statement. However, funding information should not appear in the Acknowledgments section or other areas of your manuscript. We will only publish funding information present in the Funding Statement section of the online submission form.

Response: We are sorry for this error. The funder stated in the acknowledgement section will not contribute directly to the proposed work but funds protected time for Jonathan’s PhD studies. We thought it prudent to acknowledge this funding. The funding section has been updated to include Jonathan’s PhD funders.

Academic editor review

Response: The manuscript has been formatted according to PLOS ONE’s style

 (a) Please amend your current ethics statement to state that the proposed study was been reviewed and approved by the ethics committee.

(b) Once you have amended this/these statement(s) in the Methods section of the manuscript, please add the same text to the “Ethics Statement” field of the submission form (via “Edit Submission”).

Response: The ethics statement has been revised to indicate that the protocol has already been reviewed. It has also been moved to the methods section.

 In your Methods section, please provide additional information about the participant recruitment method and the demographic details of your participants. Please ensure you have provided sufficient details to replicate the analyses such as: a) a description of how participants will be recruited and b) descriptions of where participants will be recruited and where the research will take place.

Response: The participant recruitment section has been revised to provide more detail to allow the replication of the study. It now states where and how the participants will be recruited.

 We note that you have stated that you will provide repository information for your data at acceptance. Should your manuscript be accepted for publication, we will hold it until you provide the relevant accession numbers or DOIs necessary to access your data. If you wish to make changes to your Data Availability statement, please describe these changes in your cover letter and we will update your Data Availability statement to reflect the information you provide.

Response: Regarding depositing the study data at a public online repository, we have clarified that this will be done when the study has been implemented and completed, and not when this manuscript has been accepted. This is because this is a registered protocol that has not been implemented.

 We note you have included a table to which you do not refer in the text of your manuscript. Please ensure that you refer to Table 2 in your text; if accepted, production will need this reference to link the reader to the Table.

Response: The table that had not been referenced in the text has now been referenced.

 Your ethics statement must appear in the Methods section of your manuscript. If your ethics statement is written in any section besides the Methods, please move it to the Methods section and delete it from any other section. Please also ensure that your ethics statement is included in your manuscript, as the ethics section of your online submission will not be published alongside your manuscript.

Response: The ethics statement has been moved to the methods section.

Reviewer #1: Jonathan et al, in their Protocol suggest utilization of MLR and IGRA as prognostic marker to corelate the Latent TB with HIV disease severity.

Major comments / concerns

Qn1. Author should justify the 115 number of patients, from which calculation they came to this study, would this be a clinical trials or prospective study

Response: Below is the calculation that led to the sample size of 115

Sample size was derived using the formula for diagnostic studies based on sensitivity;

n= (Z_(1-∝)^2×S_N×(1-S_N))/(L^2×prevalence) 1

SN - Anticipated sensitivity, set at 77% 2 

L2 - Absolute precision required, set at 10%

Prevalence – prevalence of LTBI in Kampala based on TST is 49%3, while the prevalence based on QuantiFeron is 65%4. The prevalence based on QauntiFeron was used.

Zα/2 - at 95% confidence level and α 0.05 is 1.96, and Zβ – at a power of 80% and β 0.2 is 0.84.

Substituting in the formula, the sample size will be 105 participants. Considering a 10% loss to follow up over the 6 months, the final sample size will be 115 participants. At this samples size, we will have at least an 80% chance of delineating the diagnostic performance of the MLR. Please see section on statistical methods under sample size and power.

Will this be a clinical trial or a prospective study?

Response: This will be a diagnostic study with a nested prospective component. Please see section on Study Methods; study design.

Qn2. I found this proposal somewhat ambitious because author propose to use monocyte to lymphocyte ratio as prognostic marker specially for latent TB. Here, my opinion would be revisit literature and revisit the concept.

Response: Thank you for this observation. We propose to explore whether the MLR can be used to monitor response to treatment for latent TB by measuring its change between baseline and three months during latent TB chemoprophylaxis. This was informed by the study by Choudhary et al5 that showed that the MLR declined with active TB treatment; before (0.41 [IQR: 0.38–0.68]) and after (0.11 [IQR: 0.048–0.348]) TB treatment. The MLR has also been used as a prognostic factor in other diseases including cardiovascular disease6 and cancer7. The complete blood count from which the MLR is derived is commonly used in routine clinical practice to monitor prognosis if infectious and other illnesses. We are therefore that our concept id based on sound scientific basis.

Qn3. Pls revise the text pertaining to the influence of IFN on proliferation of HSC. This is fundamental error, HSC require G/M-CSF or CSF-1 for proliferation and not IFN !!

Response: Thank you for this argument in reference to our hypothesis that “We therefore hypothesize that M.tb through the interferon gamma signaling effect on hematopoiesis leads to proliferation of the myeloid biased hematopoietic stem cells (HSCs) to produce target cells for propagating infection relative to the lymphoid biased HSCs leading to a higher MLR in TB infection” This hypothesis was informed by sound literature review showing that in chronic infection, IFN-γ signaling through STAT1 activates the expression of interferon regulatory factors (IRF) that can promote differentiation towards the myeloid lineage8-10. The myeloid lineage is the source of monocyte/macrophages which are the target cells for the TB bacilli. This leads to a higher MLR as seen in active TB as we have highlighted in the backgrounad. Our aim is to see whether this is replicated in latent TB.

Qn4. Monocyte population is just one rough parameter which is used in CBC. However, for the disease management, it is macrophages which are more relevant like in granuloma and in latent TB, M2 or foamy macrophage are most relevant population which promote latency and not mere monocyte. This is also due to fact that Both naive Monocyte and macrophages and CD14+/; F4/80+. During acute / active TB infection Monocyte get differentiated into CD11b+/ CD68+/Inos+ Th1 primed macrophages. during persistent / latent episode of TB , same Th1 / M1 populations, interestingly, get polarize toward M2 / Foamy macrophages and hide / carry bacilli to the gaseous granulomatous lesions of lung. So author should redefine this aspect.

Response: We appreciate the argument of the reviewer. The debate on the mononuclear phagocyte system function has not been concluded. The general agreement is that except for a few tissue macrophages like the alveolar macrophages that are derived embryonically and are capable of self-renewal at steady state11, the rest of the macrophages are derived from circulating monocytes12. The circulating monocytes replenish tissue resident macrophages especially during injury, infection and inflammation, and therefore can act as a proxy for changes in the tissue macrophages13. The reviewer seems to agree with this when he/she states that “During acute / active TB infection Monocyte get differentiated into CD11b+/ CD68+/Inos+ Th1 primed macrophages. during persistent / latent episode of TB, same Th1 / M1 populations, interestingly, get polarize toward M2 / Foamy macrophages and hide / carry bacilli to the gaseous granulomatous lesions of lung”. Therefore, the monocyte can be a fair proxy for the changes in macrophages within the tissue. The aim of our study is to evaluate a simple and easy to scale up alternative biomarker to improve the screening of latent TB more so in the resource limited settings, and not to describe the homeostasis of monocytes and tissue macrophages which has already extensively been described14. 

Qn5. Other aspect is with IGRA so I want to suggest author that author should focus on Th1/Th2 kits because other than IFN, TNF , IL-4/IL-6/IL-10/IL-13 are decisive host factors which can dictate the pathogenesis of acute and latent TB. So here author may include Th1/Th2 kit as parameters which would be much robust correlation specially when including HIV+ patients.

Response: Among the Th1/Th2 cytokines, interferon gamma is the principle activator of macrophages15, and its use in the diagnosis of latent TB is well recognized, and adopted in the World Health Organization (WHO) guidelines for the diagnosis and treatment for latent TB16, particularly in resource rich country unlike for the Th1/Th2 kits. As such it is the best available test (due to lack of a gold standard) for latent TB and therefore a suitable comparator test for any novel test for diagnosing/screening latent TB. The IGRA in the form of QuantiFeron-TB Gold plus was further improved with the inclusion of an antigen to stimulate the CD8+ T-cells, following evidence suggesting the role of the CD8+T-cells in the host defense against M.tb through producing IFN-gamma, stimulating macrophages, killing infected cells and directly lysing intracellular M.tb. The QuantiFeron-TB Gold plus is what will be used in the study as the comparator test to the MLR. Testing the role of the other members of the Th1/Th2 cytokines in diagnosing latent TB as suggested by the reviewer is beyond the intended scope of this study.

Qn6. Also for MLR , the author should use other term as in the feild of hematology / immunology MLR means mix lymphocyte ratio which is not clear here. Also I would like to suggest authors to include both M1/M2 macrophages , M1/ T cells , M1/cd4, M1/ CD8 and CD4/8 ratio as additional parameters for more in depth immunomonitoring purpose . This will give protocol strength and then analysis would be more valid.

Response: We appreciate that the MLR may carry other meanings as pointed out by the reviewer. However, in most of literature most of which is cited in this manuscript, the monocyte to lymphocyte ratio is abbreviated as MLR or ML ratio. We request that for easy comparison and identification with the contemporary literature in this area of study, we maintain the abbreviation. For better clarification and inline with good scholarly writing skills, the abbreviation is described at first use. We hope that this will remove any confusion that could have arisen with further reading down the manuscript. 

We will adopt the suggestion to use the CD4/CD8 ratio in the evaluation where it is available since CD4 are no longer routinely done except at initiation of anti-retroviral therapy, however for the M1/M2, M1/T cells M1/CD4 and M1/CD8 we feel it’s beyond the intended scope for this study and I refer the reviewer back to the response to Qn4.

Qn7. I could not follow the arguments related to Prophylactic treatment of volunteers and why only for 3 months? why Author only given INH and why not other 1st generation drug? I think it would be more appropriate to include latent TB patients which are on 2nd generation TB drug including Rapamycin and Bedaquiline drugs e.g.

Response: Thank you for this observation. The Uganda Ministry of health is in the process of updating its latent TB treatment guidelines to adopt the 3-isoniazid-rifapentine (3HP) as its first line. 3HP will be given weekly for 3 months. Therefore, it is likely that by the time the study is implemented Uganda will be using the 3HP regimen. We have updated this section to clarify that participants will be treated with 3HP for 3 months as standard of care and isoniazid as an alternative regimen. The second generation drugs suggested by the reviewer are not available in the country as treatment for latent TB.

Qn8. Did author plan to include patients with non reactive or extra pulmonary TB with PPE as well in their cohort ??. Why author did not include PPD in their analysis !!!

Response: PPD is not in regular use anymore in TB programs due to the difficulty of getting the PPD which is no longer routinely manufactured. In addition, the PPD as the skin tuberculin test (TST) use is limited by false positives due to BCG vaccination and cross-reaction with environmental non-tuberculous Mycobacteria, need for two patient visits, and false negative results due to; anergy, malnutrition, immuno-suppression and incorrect administration. These limitations are clearly laid out in the background of the manuscript. Secondly, the superiority of IGRA in terms of sensitivity and specificity particularly is the setting of extensive BCG immunization is well recognized. We are therefore confident that we will have the better comparator for MLR in IGRA than in PPD or TST.

Qn9. In the current draft, it is not clear which group of HIV patients would be included. What is the influence of Anti retro-viral therapy also contribute to IFN gamma so from that point of view, some newly diagnosed HIV patients should also be included (subjected to IEC approval but if author can include these patients, this would be baseline data. This would also address the pattern of IGRA.

Response: The current WHO policy is to test and treat all HIV patients and therefore by the end of the 3 months follow up proposed in the study, all potential participants would have been started on anti-retroviral therapy. For the first objective which is cross sectional, we will include analysis of naïve versus ART experienced participants. This has been clarified in section on study participants and statistical methods.

Minor points

Qn10. Finally, author should revise the entire document for the language because current version is difficult to analyze, e.g. second paragraph of Background !!! Please remove high impact word from the Dissemination paragraph. The author should be realistic.

Response: Thank you for this recommendation. The document has been proof read by a native English speaker. The words “high impact” have been removed. Please see the section on dissemination of study results

Qn11. Reviewer #2: This manuscript is rejected as this is a registered report protocol and it does not provide any new information at this stage. The study is a proposed hypothesis and still requires methods to be validated. However evaluation of diagnostic performance of Monocyte to Lymphocyte ratio (MLR) in comparison to IGRA assay still needs to be documented.

Outcome of this study will provide some data and insights into the comparative analysis which could be useful for development of a biomarker for LTBI diagnosis.

Response: It is unfortunate that the reviewer rejects the manuscript on the basis that it is a registered protocol. To the best of our knowledge, Plose One accepts and publishes registered protocol and it was our intention to have this published as a registered protocol. On the other hand, we are happy that the reviewer shares the same conviction that this an important concept to study and it has potential to lead to the development of a biomarker for diagnosis of LTBI. 

Qn12. Reviewer #3: In this manuscript (Registered Report Protocol) the authors proposed the diagnostic utility of the MLR (monocyte to lymphocyte ratio) in LTBI and monitoring response to preventive therapy in HIV patients.

Following are my comments on the protocol

1) In the literature available it is mentioned that Monocyte:lymphocyte ratio (M:L) has been identified as a risk factor in development of TB disease in children and those undergoing treatment for HIV in co-infected individuals. More recently, a high M:L has been shown to distinguish persons with active and latent TB from uninfected persons, and been used to predict risk of developing TB in infants. M:L has been found to reduce after treatment for HIV, and corresponded with improvement in the patient’s condition5. Therefore, in humans it appears that the M:L shows promise as an indicator of risk of developing active TB and could facilitate the targeting of preventative treatments/therapy for those who are defined as being at greater risk.

References

Naranbhai V, et al. Ratio of monocytes to lymphocytes in peripheral blood identifies adults at risk of incident tuberculosis among HIV-infected adults initiating antiretroviral therapy. J. Infect. Dis. 2014;209:500–9. doi: 10.1093/infdis/jit494.

Naranbhai V, et al. The association between the ratio of monocytes:lymphocytes at age 3 months and risk of tuberculosis (TB) in the first two years of life. BMC Med. 2014;12:120. doi: 10.1186/s12916-014-0120-7.

So, the novelty of this work is questionable. Authors are advised to revisit the literature which is full of this kind of evidences.

Response: We thank the reviewer for highlighting this literature which forms the basis of our concept in this manuscript. All the literature that the reviewer highlights above was described in the background of this manuscript. We would like to bring it to the attention of the reviewer that work in the above studies aimed to; differentiate latent from active TB, monitor response to treatment of active TB and determine incidence of active TB with a raised MLR. In our concept our focus is on the ability of the MLR to diagnose latent TB, and to the best of our knowledge this is the first attempt to show the diagnostic potential of the MLR for latent TB using IGRA (currently the best available test for latent TB) as the comparator test, particularly in HIV infected individuals with a negative screen. The symptom based screen is currently the deployed screening tool for latent TB in majority of the developing countries but is of low specificity, and therefore the need for a biomarker to improve latent TB screening. In essence we want to show if the promise seen with the MLR in active TB can be replicated in latent TB. Therefore it is our conviction that the work is novel and it will add new knowledge to the screening/diagnosis of latent TB.

Qn13. 2) Although monocyte to lymphocyte ratio is a good indicator of risk of developing active TB but role of macrophages also could not be ignored. It is advised to include macrophage variations also in this protocol.

Response: We would want to refer the reviewer to the response to Qn 4: “We appreciate the argument of the reviewer. The debate on the mononuclear phagocyte system function has not been concluded. The general agreement is that except for a few tissue macrophages like the alveolar macrophages that are derived embryonically and are capable of self-renewal at steady state11, the rest of the macrophages are derived from circulating monocytes12. The circulating monocytes replenish tissue resident macrophages especially during injury, infection and inflammation, and therefore can act as a proxy for changes in the tissue macrophages13, and therefore can act as a proxy for changes in the tissue macrophages. The aim of our study is to evaluate a simple and easy to scale up alternative biomarker to improve the screening of latent TB more so in the resource limited settings, and not to describe the homeostasis of monocytes and tissue macrophages which has extensively been described14”

Qn14. 3) Since HIV infection also manipulate the M/L ratio, authors should clarify what would be there control group.

References

The Journal of Infectious Diseases Vol. 162, No. 6 (Dec., 1990), pp. 1239-1244 Analysis of Lymphocytes, Monocytes, and Neutrophils from Human Immunodeficiency Virus (HlV)-Infected Persons for HIV DNA

Response: We thank the reviewer for sharing this article. The performance of the MLR will be measured against the IGRA positive versus the IGRA negative in a diagnostic study design. It is uncommon to have a control group in a diagnostic design. Regarding the effect of HIV, we will perform regression analysis to determine how some predictor variables affect the MLR. Viral load which is a measure of the HIV disease state will be one of the predictor variables tested. Therefore, detectable and undetectable viral load will give an indication on the effect of HIV on the performance of the MLR.

Qn15. J Infect Dis. 2014 Feb 15; 209(4): 500–509. Ratio of Monocytes to Lymphocytes in Peripheral Blood Identifies Adults at Risk of Incident Tuberculosis Among HIV-Infected Adults Initiating Antiretroviral Therapy

Response: This article is cited in our background.

Qn16. J Acquir Immune Defic Syndr. 2014 Dec 15; 67(5): 573–575. The association between the ratio of monocytes:lymphocytes and risk of tuberculosis(TB) amongst HIV infected postpartum women

Response: We appreciate this article that adds to the body of knowledge that a high MLR is associated with development of active TB. We have added this to our background. However, our focus remains to determine the diagnostic potential of the MLR in latent TB using IGRA as the comparator test. 

Qn17. 4) The authors should also clarify the HIV patients selected for the study are on ARV therapy and if these drugs are having impact on M/L ratio.

In my opinion the study need to be redesigned keeping the above mentioned points in centre.

Response: We have clarified that majority of HIV patients in the study will likely be of ARVs. However, we will recruit even those not on ARVs given that they have a negative symptom screen. Though these are likely to be few given the test and treat policy that is currently employed in the management of HIV infection/disease. This may not allow sufficient numbers to show the effect of the ARVs on the MLR. However, we know that some nucleoside transcriptase inhibitors such as zidovudine have toxic effect on the neutrophils and may suppress the bone marrow. However, the use of zidovudine has declined because tenofovir replaced it as the preferred first line and now only used in some second line regimen and therefore we may not get sufficient number of patients on zidovudine to tease out its effect on the MLR. These have been stated as limitations.

1. Zaidi SMH, Waseem HF, Ansari FA, Irfan M, S F. SAMPLE SIZE ESTIMATION OF DIAGNOSTIC TEST STUDIES IN HEALTH SCIENCES. Proc 14th International Conference on Statistical Sciences 2016 Mar; 29: 239-46.

2. La Manna MP, Orlando V, Dieli F, et al. Quantitative and qualitative profiles of circulating monocytes may help identifying tuberculosis infection and disease stages. PLOS ONE 2017 Feb 12(2): e0171358.

3. Florence N Kizza, Justin List, Allan K Nkwata, et al. Prevalence of latent tuberculosis infection and associated risk factors in an urban African setting. BMC Infectious Diseases 2015; 15: 165.

4. Biraro IA, Kimuda S, Egesa M, et al. The Use of Interferon Gamma Inducible Protein 10 as a Potential Biomarker in the Diagnosis of Latent Tuberculosis Infection in Uganda. PLoS ONE 2016; 11(1): e0146098.

5. Choudhary RK, Wall KM, Njuguna I, et al. Monocyte-to-Lymphocyte Ratio Is Associated With Tuberculosis Disease and Declines With Anti-TB Treatment in HIV-Infected Children. J Acquir Immune Defic Syndr 2019 Feb; 80(2): 174-81.

6. Ji H, Li Y, Fan Z, al e. Monocyte/lymphocyte ratio predicts the severity of coronary artery disease: a syntax score assessment. BMC Cardiovasc Disord 2017 2017; 90.

7. Gong J, Jiang H, Shu C, al e. Prognostic value of lymphocyte-to-monocyte ratio in ovarian cancer: a meta-analysis. J Ovarian Res 2019; 12: 51.

8. de Bruin AM, Libregts SF, Valkhof M, Boon L, Touw IP, MA N. IFNγ induces monopoiesis and inhibits neutrophil development during inflammation. Blood 2012; 119: 1543-54.

9. Morales-Mantilla DE, KY K. The Role of Interferon-Gamma in Hematopoietic Stem Cell Development, Homeostasis, and Disease. Curr Stem Cell Rep 2018; 4(3): 264-71.

10. Baldridge MT, King KY, Boles NC, al e. Quiescent haematopoietic stem cells are activated by IFN-γ in response to chronic infection. Nature 2010 June; 465: 793-7.

11. Epelman S, Lavine KJ, Randolph GJ, al e. Origin and Functions of Tissue Macrophages

Immunity 2014 Jul 41(1): 21-35.

12. Hume DA, Irvine KM, Pridans C. The Mononuclear Phagocyte System: The Relationship between Monocytes and Macrophages Trends in Immunology 2019 Feb; 40(2).

13. Zhao Y, Zou W, Du J, al e. The origins and homeostasis of monocytes and tissue‐resident macrophages in physiological situation. Cellular Physiology 2018 Jan; 233(10): 6425-39.

14. Ginhoux F, Jung S, al e. Monocytes and macrophages: developmental pathways and tissue homeostasis. Nat Rev Immunol 2014; 14: 392-404.

15. Teixeira LK, Fonseca BPF, Barboza BA, JPB V. The role of interferon-g on immune and allergic responses. Mem Inst Oswaldo Cruz, Rio de Janeiro, 2005; 100(Suppl.I): 137-44.

16. 2018 W. Updated and consolidated guidelines for programmatic management of latent tuberculosis infection.

---

## [Editor Report · Decision Letter 1]

21 Oct 2020

Utility of the monocyte to lymphocyte ratio in diagnosing latent tuberculosis among HIV-infected individuals with a negative tuberculosis symptom screen

PONE-D-20-22616R1

Dear Dr. Mayito,

We’re pleased to inform you that your manuscript has been judged scientifically suitable for publication and will be formally accepted for publication once it meets all outstanding technical requirements.

Kind regards,

Seyed Ehtesham Hasnain

Academic Editor

PLOS ONE

Additional Editor Comments (optional):

I have gone through the revised manuscript and also the Authors response to the comments of the reviewers. The manuscript was sent for revision and Authors have modified the manuscript keeping in mind the comments of the Reviewers. The issue of Data Repository needs to be clarified at the PLoS One level. In my view, the authors have otherwise satisfactorily addressed all the comments made by the reviewers and added all required information, and have revised the manuscript accordingly. I recommend this manuscript for publication.
---

## [Editor Report · Acceptance letter]

29 Oct 2020

PONE-D-20-22616R1 

Utility of the monocyte to lymphocyte ratio in diagnosing latent tuberculosis among HIV-infected individuals with a negative tuberculosis symptom screen 

Dear Dr. Mayito:

I'm pleased to inform you that your manuscript has been deemed suitable for publication in PLOS ONE. Congratulations! Your manuscript is now with our production department. 

Kind regards, 

on behalf of

Prof Seyed Ehtesham Hasnain 

Academic Editor

PLOS ONE